# Parameters Design and Optimization of SiC MOSFET Driving Circuit with Consideration of Comprehensive Loss and Voltage Stress

**DOI:** 10.3390/mi14030505

**Published:** 2023-02-21

**Authors:** Haihong Qin, Zhenhua Ba, Sixuan Xie, Zimo Zhang, Wenming Chen, Qian Xun

**Affiliations:** 1Department of Electrical Engineering, the Center for More-Electric-Aircraft Power System, Nanjing University of Aeronautics and Astronautics, Nanjing 211106, China; 2Department of Electrical Engineering, Chalmers University of Technology, 41279 Gothenburg, Sweden

**Keywords:** driving circuit, parameters optimization, gate-source voltage, SiC MOSFET

## Abstract

In conventional parameters design, the driving circuit is usually simplified as an RLC second-order circuit, and the switching characteristics are optimized by selecting parameters, but the influence of switching characteristics on the driving circuit is not considered. In this paper, the insight mechanism for the gate-source voltage changed by overshoot and ringing caused by the high switching speed of SiC MOSFET is highlighted, and we propose an optimized design method to obtain optimal parameters of the SiC MOSFET driving circuit with consideration of parasitic parameters. Based on the double-pulse circuit, we evaluated the influence of main parameters on the gate-source voltage, including driving voltage, driving resistance, gate parasitic inductance, and stray inductance of the power circuit. A SiC-based boost PFC is constructed and tested. The test results show that the switching loss can be reduced by 7.282 W by using the proposed parameter optimization method, and the over-voltage stress of SiC MOSFET is avoided.

## 1. Introduction

To reduce carbon emissions, it is crucial to improve the efficiency of motor drives to promote the development of electric vehicles, new energy power generation, and other industries [1,2]. As a typical wide bandgap (WBG) device, silicon carbide (SiC) metal-oxide-semiconductor field-effect-transistor (MOSFET) shows great advantages over silicon (Si) MOSFET in terms of on-resistance, switching speed, and thermostability [3,4,5]. The replacement of Si MOSFETs with SiC MOSFETs can improve the efficiency and power density of power electronics and promote the development of motor drives [6]. However, the high switching speed of SiC MOSFET makes it very sensitive to parasitic parameters in the circuit [7,8], and the voltage and current are susceptible to producing overshoot and oscillation [9,10]. Also, this increases the electrical stress of the device, resulting in accelerated aging and even failure of the device. To ensure the safety of SiC MOSFET devices, designing an efficient and reliable driving circuit is necessary and becomes an urgent task.

At present, for the design of driving circuit parameters of SiC MOSFET, researchers have noticed the characteristics of SiC MOSFET and made special consideration from the aspects of driving voltage setting, driving chip current capacity, rise/fall time, PCB layout, and so on. SiC device manufacturers provide recommended driving voltage in their technical manuals, but these values are simply estimated and are not considered in conjunction with other driving parameters of the actual driving circuit. In ref. [11], the effects of driving resistance and parasitic capacitance of SiC MOSFET on the maximum turn-on speed are studied. However, turn-off switching characteristic analysis and driving resistance selection guidance methods are not given. In ref. [12], the influence of different driving voltage, driving resistance, and gate-source capacitors on the switching characteristics of SiC MOSFET are analyzed to suppress the oscillation and overshoot, but this method will increase the switching time of the device. In ref. [13], the appropriate driving resistor is selected by comprehensively considering the switching loss and temperature rise. In ref. [14], the RLC response of the driving circuit is analyzed, and the parasitic inductance optimization method is given considering the nonlinear characteristics of the capacitor. Reference [15] studies the influence of driving voltage and driving resistance on suppressing gate oscillation through the loss change and damping effect. Among these methods, the coupling effects of the power circuit are not considered, which may cause the actual gate-source voltage to exceed the design value.

However, due to the high d*v*/d*t* and d*i*/d*t* of SiC MOSFET, its interaction with circuit parasitic parameters will produce obvious voltage and current oscillation. In ref. [16], the influence of circuit parasitic parameters on gate-source voltage is mainly discussed. In ref. [17], the RLC second-order equivalent circuit model is proposed for the turn-on and turn-off of SiC MOSFET. A simple mathematical formula is derived, which provides a theoretical analysis basis for the study of the switching oscillation phenomenon of SiC MOSFET and has important guiding significance for the design of buffer or damping circuits. During the actual testing, we found that the overshoot and ringing of the power circuit will have a great impact on the gate-source voltage and increase the gate voltage stress. However, this phenomenon is often mistaken for the second-order oscillation of the driving circuit itself. Therefore, the influence of high-frequency oscillation on the gate-source voltage must be considered when designing the driving circuit parameters of SiC MOSFET. Since the dynamic characteristics of SiC devices are closely related to stray parameters in the circuit, a method to extract stray inductance and capacitance of the power circuit is proposed [18]. In ref. [19], the gate oscillation caused by d*v*/d*t* and d*i*/d*t* feedback is analyzed. It is proposed to increase the driving resistance and parallel gate-source capacitance to suppress the oscillation, but the selection guideline of driving resistance is not given. In ref. [20], the switching dynamic characteristic analysis model is proposed according to the datasheet and the parasitic effect of the external circuit, but the dynamic characteristic of gate-source voltage is not analyzed. Moreover, due to the additional loss caused by the oscillation peak, the switching loss will also increase. Therefore, the influence of oscillation loss should be considered in the loss calculation to make the calculation results more accurate [21].

In this paper, the mechanism of voltage and current ringing coupled to gate-source voltage is analyzed, the relevant mathematical model is established, and the parameters optimization design method is proposed. This method can reduce the switching loss and conduction loss as much as possible and ensure gate reliability. The rest of the paper is organized as follows. In Section 2, the mathematical model of gate-source voltage considering the coupling relationship of the power circuit is developed. In Section 3, based on the consideration of balancing comprehensive loss and overvoltage stress, the parameter-optimized design method for the driving circuit is proposed. Section 4 gives the experimental results and Section 5 concludes the work.

## 2. Modeling of Transient Gate-Source Voltage

The SiC MOSFET double pulse test circuit is shown in Figure 1 where the main parasitic parameters are also considered. *V*_DC_ is the bus voltage, *I*_l_ is the load current, and *C*_L_ is the parasitic capacitance of the load inductor. D_H_ is the ideal SiC SBD and *C*_J_ is the equivalent junction capacitance of SiC SBD. *C*_GS_, *C*_GD,_ and *C*_DS_ are the gate-source capacitance, gate-drain capacitance, and drain-source capacitance of SiC MOSFET respectively, *L*_D(int)_ and *L*_S(int)_ are parasitic inductance introduced by drain and source pins in SiC MOSFET package respectively, *R*_G(int)_ is the gate internal resistance of SiC MOSFET, and *R*_G(ext)_ is the external driving resistance, *L*_G_ is the parasitic inductance of the driving circuit. *L*_D(ext)_ and *R*_loop_ are the equivalent parasitic inductance and stray resistance of the PCB wiring between the positive terminal of the DC bus and the drain of SiC MOSFET respectively, *L*_S(ext)_ is the parasitic inductance of the line between SiC MOSFET source and ground.

The double pulse test circuit is used to develop the mathematical model of gate-source voltage. Due to parasitic parameters, the switching process of the SiC MOSFET is shown in Figure 2.

The turn-on process can be divided into five stages according to the change of current and voltage, which is described below:Stage 1 [*t*_0_~*t*_1_]

At the time instance *t*_0_, the input capacitance *C*_ISS_ starts to charge, and the gate-source voltage *v*_GS_ rises, whereas, the drain current *i*_D_ and drain-source voltage *v*_DS_ do not change. The gate-source voltage at this stage can be expressed as:(1)$$v_{\mathrm{GS}}=\frac{1}{L_{\mathrm{GS}}C_{\mathrm{ISS}}s^{2}+R_{G}C_{\mathrm{ISS}}s+1}\times \frac{V_{\mathrm{DRV}}}{s}$$
where the input capacitance follows *C*_ISS_ = *C*_GS_ + *C*_GD_, the gate-source inductance follows *L*_GS_ = *L*_G_ + *L*_S(int)_, and the driving resistance follows *R*_G_ = *R*_G(int)_ + *R*_G(ext)_. The driving voltage is *V*_DRV_, which is equivalent to step excitation in the zero-state response of the driving circuit.

According to (1), the driving circuit parameters are the main factors affecting the gate-source voltage *v*_GS_.

2.Stage 2 [*t*_1_~*t*_2_]

At the time instance *t*_1_, the gate-source voltage reaches the threshold voltage, the channel begins to turn on and the drain current *i*_D_ gradually rises. Due to the small d*v*_DS_/d*t*, the current flowing through the parasitic capacitance of SiC MOSFET is also small, the channel current *i*_CH_ can be approximately regarded as the drain current, which can be expressed as:(2)$$v_{\mathrm{GS}}\left(t\right)=\frac{i_{D}\left(t\right)}{g_{\mathrm{fs}}}+V_{\mathrm{TH}}$$
(3)$$i_{D}=\frac{\omega_{0(2)}^{2}g_{\mathrm{fs}}\left(V_{\mathrm{DRV}}-V_{\mathrm{TH}}\right)}{s^{2}+2\delta_{\left(2\right)}s+\omega_{0(2)}^{2}}$$
where
(4)$$\left\{\begin{array}{c} \delta_{\left(2\right)}=\frac{R_{G}\left(C_{\mathrm{GS}}+C_{\mathrm{GD}}+g_{\mathrm{fs}}R_{\mathrm{loop}}C_{\mathrm{GD}}\right)+g_{\mathrm{fs}}L_{S(\mathrm{int})}}{2g_{\mathrm{fs}}R_{G}L_{\mathrm{stray}}C_{\mathrm{GD}}}\\ \omega_{0(2)}=\frac{1}{\sqrt{g_{\mathrm{fs}}R_{G}L_{\mathrm{stray}}C_{\mathrm{GD}}}} \end{array}\right.$$

The gate-source voltage increases from the threshold voltage to the Miller voltage with the drain current increasing. It can be seen from (4) that the factors affecting the gate-source voltage include the parasitic capacitance and transfer characteristics of SiC MOSFET, driving resistance, stray parameters of the power circuit, and working conditions.

3.Stage 3 [*t*_2_~*t*_3_]

At the time instance *t*_2_, the drain current *i*_D_ increases to the load current *I*_L_, and the current of SiC SBD decreases to zero. At this time, the parasitic capacitances of the power circuit (*C*_J_ and *C*_L_) are charged by the reverse voltage, and the drain current *i*_D_ spikes and causes high-frequency oscillation. This stage is in the Miller platform stage, which can be subdivided into two stages [*t*_2_~*t*_P_] and [*t*_P_~*t*_3_] according to the changes in drain current and drain-source voltage. More details are shown below.


(a)[*t*_2_~*t*_P_]


The drain current *i*_D_ begins to overshoot at the time instance *t*_2_ and reaches the current peak *I*_peak_ at the time instance *t*_P_, while the drain current change rate d*i*_D_/d*t* decreases to zero. Due to the change of *i*_D_, the gate-source voltage *v*_GS_ starts to rise from Miller voltage *V*_P_, and the peak of Miller platform voltage at *t*_P_ can be expressed as:(5)$$v_{\mathrm{GS}}(t_{P})=\frac{I_{\mathrm{peak}}}{g_{\mathrm{fs}}}+V_{\mathrm{TH}}$$

The drain current *i*_D_ can be expressed as:(6)$$i_{D}=\frac{L_{\mathrm{stray}}{\left.\frac{di_{D}\left(t\right)}{dt}\right|}_{t=t_{2}}}{L_{\mathrm{stray}}\left(C_{J}+C_{L}\right)s^{2}+R_{\mathrm{loop}}\left(C_{J}+C_{L}\right)s+1}$$

Since the stray resistance, *R*_loop_, is very small and follows *δ*_(31)_^2^ < *ω*_0(31)_, the power circuit works in the underdamped state. From (6), it can be seen that the current peak *I*_peak_ is related to the switching speed and stray parameters of the power circuit. These factors will also affect the voltage peak of the gate-source voltage *v*_GS_ at this stage.


(b)[*t*_P_~*t*_3_]


At the time instance *t*_P_, the drain current *i*_D_ begins to decrease and the drain-source voltage *v*_DS_ also decreases. Due to the large change rate of drain-source voltage d*v*_DS_/d*t*, the displacement current on the parasitic capacitance of SiC MOSFET cannot be ignored. Therefore, the channel current *i*_CH_ is no longer approximate to the drain current, which can be expressed as:(7)$$i_{\mathrm{CH}}\left(t\right)+g_{\mathrm{fs}}\Delta v_{\mathrm{GS}}\left(t\right)=g_{\mathrm{fs}}\left(V_{\mathrm{DRV}}-V_{\mathrm{TH}}\right)$$
(8)$$v_{\mathrm{GS}}\left(t\right)=\frac{i_{\mathrm{CH}}\left(t\right)}{g_{\mathrm{fs}}}+V_{\mathrm{TH}}$$
(9)$$i_{D}=\frac{\frac{g_{\mathrm{fs}}\left(V_{\mathrm{DRV}}-V_{\mathrm{TH}}\right)C_{H}+\left(C_{\mathrm{OSS}}+g_{\mathrm{fs}}R_{G}C_{\mathrm{GD}}\right)I_{L}}{C_{H}\left(C_{\mathrm{OSS}}+g_{\mathrm{fs}}R_{G}C_{\mathrm{GD}}\right)}}{L_{\mathrm{stray}}s^{2}+\left(R_{\mathrm{loop}}+\frac{g_{\mathrm{fs}}L_{S(\mathrm{int})}}{C_{\mathrm{OSS}}+g_{\mathrm{fs}}R_{G}C_{\mathrm{GD}}}\right)s+1}$$
where
(10)$$\left\{\begin{array}{c} \delta_{\left(32\right)}=\frac{R_{\mathrm{loop}}+\frac{g_{\mathrm{fs}}L_{S(\mathrm{int})}}{C_{\mathrm{OSS}}+g_{\mathrm{fs}}R_{G}C_{\mathrm{GD}}}}{2L_{\mathrm{stray}}}\\ \omega_{0(32)}=\sqrt{\frac{C_{H}+C_{\mathrm{OSS}}+g_{\mathrm{fs}}R_{G}C_{\mathrm{GD}}}{L_{\mathrm{stray}}C_{H}\left(C_{\mathrm{OSS}}+g_{\mathrm{fs}}R_{G}C_{\mathrm{GD}}\right)}} \end{array}\right.$$
where the output capacitance follows *C*_OSS_ = *C*_GD_ + *C*_DS_, and the equivalent parasitic capacitance follows *C*_H_ = *C*_J_ + *C*_L_.

According to (7)~(9), the gate-source voltage is related to the drain current at this time. It can be seen that there is an oscillation component in the drain current, which is related to the parasitic capacitance and transfer characteristics of SiC MOSFET, driving circuit parameters, power circuit stray parameters, and working conditions.

4.Stage 4 [*t*_3_~*t*_4_]

At the time instance *t*_3_, the drain-source voltage *v*_DS_ drops to *V*_DS(ON)_. At this time, the drain current *i*_D_ can be expressed as:(11)$$i_{D}=\frac{I_{L}}{L_{\mathrm{stray}}\left(C_{J}+C_{L}\right)s^{2}+R_{\mathrm{loop}}\left(C_{J}+C_{L}\right)s+1}$$

The general solution of drain current *i*_D_ can be expressed as:(12)$$i_{D}\left(t\right)=I_{L}+e^{-\delta_{\left(4\right)}\left(t-t_{3}\right)}\left\{K_{1(4)}\cos\left[\omega_{\left(4\right)}\left(t-t_{3}\right)\right]+\right.\left.K_{2(4)}\sin\left[\omega_{\left(4\right)}\left(t-t_{3}\right)\right]\right\}$$
where
(13)$$\left\{\begin{array}{c} K_{1(4)}=i_{D}\left(t_{3}\right)-I_{L}\\ K_{2(4)}=\frac{{\left.\frac{di_{D}\left(t\right)}{dt}\right|}_{t=t_{3}}+\delta_{\left(4\right)}K_{1(4)}}{\omega_{\left(4\right)}} \end{array}\right.$$

The gate-source voltage *v*_GS_ continues to rise from Miller voltage *V*_P_. At the same time, the current ringing senses the voltage ringing on the common source parasitic inductance *L*_S(int)_ and is coupled to the gate circuit to become an excitation source, so that the gate-source voltage *v*_GS_ superimposes high-frequency oscillation.
(14)$$\Delta v_{\mathrm{GS}}=\frac{K_{3(4)}\left[s\sin\psi +\omega_{\left(4\right)}\cos\psi \right]}{\left[{\left(s+\delta_{\left(1\right)}\right)}^{2}+\omega_{\left(1\right)}{}^{2}\right]\left[{\left(s+\delta_{\left(4\right)}\right)}^{2}+\omega_{\left(4\right)}{}^{2}\right]}$$
where
(15)$$\left\{\begin{array}{c} K_{1(4)}=i_{D}\left(t_{3}\right)-I_{L}\\ K_{2(4)}=\frac{{\left.\frac{di_{D}\left(t\right)}{dt}\right|}_{t=t_{3}}+\delta_{\left(4\right)}K_{1(4)}}{\omega_{\left(4\right)}} \end{array}\right.$$

Then the expression of gate-source voltage *v*_GS_ at this time is:
(16)$$v_{\mathrm{GS}}\left(t\right)=V_{\mathrm{DRV}}\left\{1+2\left|K_{4(4)}\right|e^{-\delta_{\left(1\right)}\left(t-t_{3}\right)}\cdot \right.\left.\cos\left[\omega_{\left(1\right)}\left(t-t_{3}\right)+\theta \right]\right\}+\Delta v_{\mathrm{GS}}\left(t\right)$$
where
(17)$${\left.\frac{\frac{V_{\mathrm{DRV}}}{L_{\mathrm{GS}}C_{\mathrm{GS}}}\cdot \left(s+\delta_{\left(1\right)}-j\omega_{\left(1\right)}\right)}{s\cdot \left[{\left(s+\delta_{\left(1\right)}\right)}^{2}+\omega_{\left(1\right)}{}^{2}\right]}\right|}_{s=-\delta +j\omega }=\left|K_{4(4)}\right|e^{j\theta }$$

It can be seen from (16) that in addition to the second-order oscillation caused by the driving circuit, the gate-source voltage will also superimpose the high-frequency oscillation from the power circuit. The factors affecting the gate-source voltage include driving circuit parameters, power circuit parameters, and working conditions.

5.Stage 5 [*t*_4_~*t*_5_]

At the time instance *t*_4_, the gate-source voltage rises to *V*_DRV_, and then the gate-source voltage spike and attenuation oscillation appear. The gate-source voltage at this stage is:
(18)$$v_{\mathrm{GS}}\left(t\right)=V_{\mathrm{DRV}}\{1+2\left|K_{\left(5\right)}\right|e^{-\delta_{\left(1\right)}\left(t-t_{4}\right)}\cdot \cos\left[\omega_{\left(1\right)}\left(t-t_{4}\right)+\theta \right]\}+\Delta v_{\mathrm{GS}}\left(t\right)$$
where *K*_(5)_ = *K*_4(4)_, and the definition of other parameters is the same as that in stage 4.

The generation mechanism of ∆*v*_GS_ in the turn-off process is similar to that in the turn-on process, which will not be repeated. According to the mathematical model, when considering the influence of switching ringing, the driving circuit cannot be simply equivalent to an RLC circuit and the gate-source voltage will superimpose a high-frequency oscillation voltage ∆*v*_GS_, which is related to the switching speed and power circuit parameters. ∆*v*_GS_ will increase the overshoot and oscillation amplitude of gate-source voltage. Therefore, the influence of d*i*/d*t* and stray inductance of the power circuit must be considered when designing driving parameters.

## 3. Parameter Optimized Design Method

We propose an optimized design method for driving parameters considering the influence of parasitic parameters in the power circuit, as shown in Figure 3. The methodology is divided into three steps to illustrate the process of selecting the optimal driving parameters. The main steps are described as follows.

Step I: Since the gate-source inductance and stray inductance are the main factors causing gate oscillation, the PCB layout should be optimized as much as possible to make the parasitic inductance less than the recommended value.

Step II: To ensure gate reliability, there is a margin between the maximum allowable gate-source voltage *v*_GS(max)_ and the gate-source withstand voltage *V*_GSS_. Then the driving parameters combination (*V*_DRV_, *R*_G_) is calculated according to the *v*_GS(max)_.

Step III: Since the switching loss and conduction loss of the power transistor is different under different driving parameters, the driving circuit parameters should be determined according to the principle of optimal comprehensive loss, which means the combination of switching loss and conduction loss for the device approaches minimum under this set of driving parameter.

This parameter optimization design method fully considers the influence of ringing caused by parasitic parameters on gate-source voltage and ensures the gate reliability of SiC MOSFET by optimizing stray inductance without affecting the switching speed as much as possible.

### 3.1. Parasitic Parameters Design of Power Circuit

To reduce the turn-on current spike, the equivalent junction capacitances *C*_J_ and *C*_L_ should be as small as possible. Therefore, SCS240AE2 (SiC SBD, Rohm) and an air-core inductor are selected. Also, to reduce the stray loss, the stray resistance *R*_loop_ should be as small as possible.

When designing the stray inductance *L*_stray_, for different stray inductances, the maximum gate-source voltage *v*_GS(max)_ is limited by dynamically changing the gate-source inductance *L*_GS_ and the driving resistance *R*_G_. At the same time, the switching energy loss and device stress are paid attention to, and the acceptable design range of stray inductance *L*_stray_ is obtained. The specific parameters and experimental test data are shown in Table 1, and SiC MOSFET SCT3060AL (650 V/39 A, Rohm) used for simulation has 12 Ω internal gate resistance.

In the simulation, the bus voltage *V*_DC_ is 400 V and the load current *I*_L_ is 20 A. The maximum voltage *v*_GS (max)_ is limited to 21 V when the gate-source inductance *L*_GS_ is 40 nH and 20 nH, respectively. Figure 4 shows the switching processes of SiC MOSFET with various stray inductances.

Under the condition that *V*_DRV_ is 18 V and *R*_G_ is 17 Ω, the stray inductance *L*_stray_ is designed with the following constraints. It cannot exceed 60 nH when the gate-source inductance *L*_GS_ is 40 nH and cannot exceed 125 nH when *L*_GS_ is 20 nH. The larger gate-source inductance *L*_GS_ causes the larger gate-source voltage oscillation under the same *V*_DRV_ and *R*_G_, then the acceptable stray inductance *L*_stray_ will be smaller.

When the gate-source inductance *L*_GS_ is 40 nH and 20 nH respectively, the maximum stray inductance *L*_stray_ limited by the voltage stress is reduced from 260 nH to 125 nH. The smaller gate-source inductance *L*_GS_ allows a faster switching speed under the same *V*_DRV_ and *R*_G_. This results in the increase of the turn-off voltage spike and the decreasing acceptable range of stray inductance *L*_stray_.

Figure 5 shows the influence of different gate-source inductance *L*_GS_ and stray inductance *L*_stray_ on the switching energy loss of the device. The larger *L*_stray_ results in the larger driving resistance *R*_G_ to suppress the gate-source voltage oscillation leading to a slower switching speed. During the turn-on process, it can be seen from (9) that the larger *L*_stray_ gives the smaller voltage platform in stage 2 and the turn-on loss is reduced. In summary, the turn-on energy loss decreases first and then increases, and the turn-off energy loss increases due to the increase of *R*_G_.

According to the above analysis and considering the voltage stress, switching energy loss, and physical space limitation of the circuit, the stray inductance *L*_stray_ should not exceed 60 nH.

### 3.2. Parasitic Parameters Design of Driving Circuit

Under the optimized parasitic parameters of the power circuit, the gate-source voltage oscillation and voltage stress can be optimized. On this basis, by limiting the maximum gate-source voltage *v*_GS(max)_ to 21 V and adjusting the parameters of the driving circuit, the switching loss and conduction loss of the device can be effectively reduced, and the driving parameters can be optimized with respect to the optimal comprehensive loss. When *L*_stray_ is 60 nH and *v*_GS(max)_ is 21 V, the specific driving circuit parameters and test data are shown in Table 2.

Figure 6 shows the switching processes under different gate-source inductance *L*_GS_, driving resistance *R*_G,_ and driving voltage *V*_DRV_. When *R*_G_ increases, *V*_DRV_ will also increase at the same time under the same gate-source inductance *L*_GS_, and the turn-on current peak increases slightly. Since the positive driving voltage does not affect the turn-off process, the turn-off voltage spike decreases. Under the same driving resistance *R*_G_, the smaller *L*_GS_ gives the higher *V*_DRV_, which makes the turn-on current peak decreases slightly. Since the positive driving voltage does not affect the turn-off process, the turn-off voltage spike decreases.

Figure 7 shows the effects of different driving parameters *L*_GS_, *R*_G,_ and *V*_DRV_ on the comprehensive loss of the device. When *R*_G_ increases and *V*_DRV_ also increases under the same *L*_GS_, the turn-on energy loss *E*_on_ decreases firstly and then increases, and the conduction loss decreases. Since the positive driving voltage does not affect the turn-off process and the turn-off loss continues to increase, the comprehensive loss first decreases and then increases. Under the same *R*_G_, the smaller *L*_GS_ is, the higher *V*_DRV_ can be, and the turn-on loss and conduction loss are reduced. Since the positive driving voltage does not affect the turn-off loss, thus the comprehensive loss is reduced.

According to the principle of optimal comprehensive loss, when *L*_GS_ is 40 nH, the optimized value of *R*_G_ is 15 Ω, and the damping ratio is 1.09. When *L*_GS_ is 20 nH, the optimized value of *R*_G_ is 13 Ω, and the damping ratio is 1.34. It can be seen that the smaller the gate-source inductance *L*_GS_, the greater the damping ratio of the optimal driving parameters. This is because the decrease of *L*_GS_ increases the switching speed, resulting in more serious high-frequency oscillation and higher damping is required to suppress this oscillation.

## 4. Experimental Verification and Discussion

As shown in Figure 8, under the same driving circuit parameters, when SiC MOSFET is not connected to the power circuit, the gate-source voltage overshoot is 1.7 V and the steady-state recovery time is 125 ns. When SiC MOSFET operates at 400 V/20 A, the gate-source voltage overshoot increases to 3.8 V and the steady-state recovery time increases to 300 ns. This means that the ringing of the power circuit will be coupled to the driving circuit.

To evaluate the influence of different driving parameter combinations (*V*_DRV_, *R*_G_) on the comprehensive loss and voltage stress of SiC MOSFET (SCT3060AL, Rohm, Kyoto, Japan), the switching characteristics of SiC MOSFET are tested on a double pulse test circuit and a multi-pulse test circuit, as shown in Figure 9. The test conditions are described in Table 3.

### 4.1. Double Pulse Test

The switching waveform of SiC MOSFET under different driving parameter combinations is shown in Figure 10. Different combinations of driving parameters are selected to limit the maximum gate-source voltage *v*_GS(max)_ to 21 V. Obviously, with the decrease of *R*_G_, the oscillation amplitude of the gate-source voltage *v*_GS_ will increase. As *R*_G_ decreases from 22 Ω to 12 Ω, the voltage *v*_GS_ overshoot increases from 1.2 V to 7.2 V. *R*_G_ decreases while *V*_DRV_ decreases, and the turn-on current stress decreases from 24.5 A to 23.0 A. The positive driving voltage has little effect on the turn-off process, the turn-off voltage peak increases from 485 V to 595 V with the decrease of *R*_G_, which increases by 22.68%.

Figure 11 shows the switching energy loss and comprehensive loss of SiC MOSFET under different driving parameter combinations, in which the switching energy loss is obtained by integrating the intersection part of voltage and current waveform of an oscilloscope, and the comprehensive loss includes switching loss and conduction loss, switching loss is obtained by multiplying the switching energy loss by the switching frequency. As *R*_G_ decreases, the gate-source voltage overshoot increases, resulting in limited *V*_DRV_ and increased turn-on energy loss. When the *R*_G_ decreases from 22 Ω to 12 Ω, the turn-on energy loss increases from 153.8 μJ to 217.2 μJ, which increased by 41.22%. The positive driving voltage has little effect on the turn-off process. Therefore, the smaller *R*_G_ gives the smaller the turn-off energy loss. When the *R*_G_ decreases from 22 Ω to 12 Ω, the turn-off energy decreases by 35.59%. According to the comprehensive loss under different switching frequencies, it can be seen that compared with the combination of (15.2 V, 12.0 Ω), the comprehensive loss of (22.0 V, 22.0 Ω) is reduced by 30.175 W at 200 kHz and 49.89 W at 600 kHz. With the increase in switching frequency, the combination of driving parameters with small switching energy loss shows better performance.

Based on the principle of optimal comprehensive loss, when *L*_GS_ is 42.52 nH, we select (18.8 V, 18.0 Ω) as the best combination of driving parameters (*V*_DRV_, *R*_G_).

### 4.2. Multi Pulse Test

To further verify the effectiveness of the proposed method, the experiment is conducted on a boost PFC converter through a multi-pulse test. The gate-source inductance *L*_GS_ is optimized below 20 nH. The multi-pulse waveforms and switching waveforms under different driving parameter combinations are shown in Figure 12. It can be seen that when the gate-source inductance *L*_GS_ is 20 nH, the driving circuit itself does not oscillate even if the driving resistance *R*_G_ is 12 Ω. At this time, the oscillation of gate-source voltage is only caused by the high-frequency oscillation of the power circuit. The influence of driving parameter combination on switching characteristics decreases with the decrease of gate-source inductance. As the driving resistance *R*_G_ decreases, the switching speed increases slightly, and the high-frequency oscillation amplitude of the gate-source voltage also increases slightly.

The switching energy loss of SiC MOSFET and the efficiency of Boost PFC converter under different driving parameter combinations are shown in Figure 13. Similar to the double pulse experimental results, the total switching energy loss of SiC MOSFET also decreases first and then increases. The efficiency curve of Boost PFC corresponds to the switching energy loss of SiC MOSFET. In the optimized six groups of driving parameter combinations, the maximum difference in switching energy loss is 36.41 μJ, the maximum difference in switching loss is 7.282 W and the maximum difference in efficiency is 0.086%.

Figure 14 shows the total loss distribution of the Boost PFC converter at the switching frequency of 200 kHz. The loss of each part of the Boost PFC converter is calculated according to its commonly used loss model [22]. The switching loss of SiC MOSFET accounts for 30.7% of the total loss. Because the gate-source inductance *L*_GS_ is optimized and six groups of driving parameter combinations selected in the experiment are optimized, the change in efficiency is not obvious. The efficiency difference is calculated according to the switching energy loss difference and switching frequency, and the results are consistent with the experiment.

Based on the consideration of optimal comprehensive loss, when *L*_GS_ is 19.34 nH, select (20.0 V, 18.0 Ω) as the best combination of driving parameters. According to the previous analysis, if *L*_GS_ can continue to be optimized, *V*_DRV_ can be further increased under the same driving resistance and the comprehensive loss will be further reduced.

## 5. Conclusions

The ringing in the power circuit will be coupled to the driving circuit through the junction capacitance and common source parasitic inductance of the SiC MOSFET, so the gate-source voltage superimposes the high-frequency oscillation. Based on the parasitic parameters of SiC MOSFET, a mathematical model of gate-source voltage considering the influence of power circuit ringing is established in this paper. Then based on the model, a methodology is proposed to optimize the parameters of the SiC driving circuit with respect to an optimal comprehensive loss of SiC MOSFET and ensure its reliable electrical stress. Through comprehensive experiments, the reliability and comprehensive loss of the driving circuit are taken as evaluation indexes, this paper shows the influence mechanism of different stray inductances and driving circuit parameters on overshoot and ringing of gate-source voltage, gives the best driving parameters combination under different gate-source parasitic inductances.

The parameters optimization method consists of three main steps. In the first step, through the compact design of the circuit layout, the parasitic inductance in the circuit will be optimized below the recommended value, which will reduce the high-frequency oscillation and gate oscillation of SiC MOSFET, and optimize other driving parameters to reduce the comprehensive loss. In the second step, the combinations of driving parameters are optimized in a narrow range. In the third step, the optimal driving parameter combination is identified, which optimizes the comprehensive loss of SiC MOSFET under the condition of ensuring voltage stress. Using this method, the optimizing driving parameters can be obtained, and the switching loss is reduced by 7.28 W in a 200 kHz Boost PFC converter.

## Figures and Tables

**Figure 1 micromachines-14-00505-f001:**
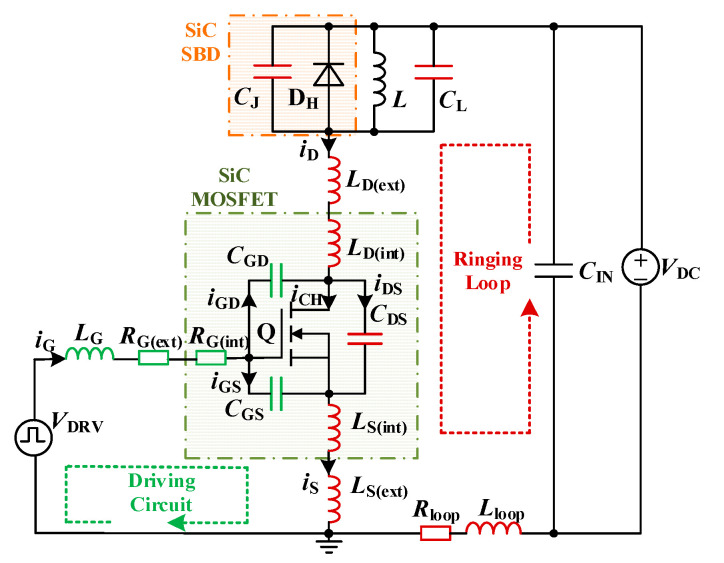
Double pulse circuit considering parasitic parameters for SiC MOSFET.

**Figure 2 micromachines-14-00505-f002:**
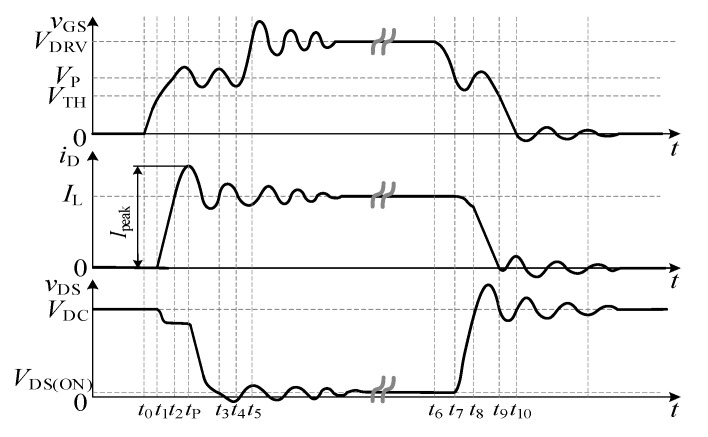
Switching waveform of SiC MOSFET considering parasitic parameters. From top to bottom: gate-source voltage *v*_GS_, drain current *i*_D,_ and drain-source voltage *v*_DS_.

**Figure 3 micromachines-14-00505-f003:**
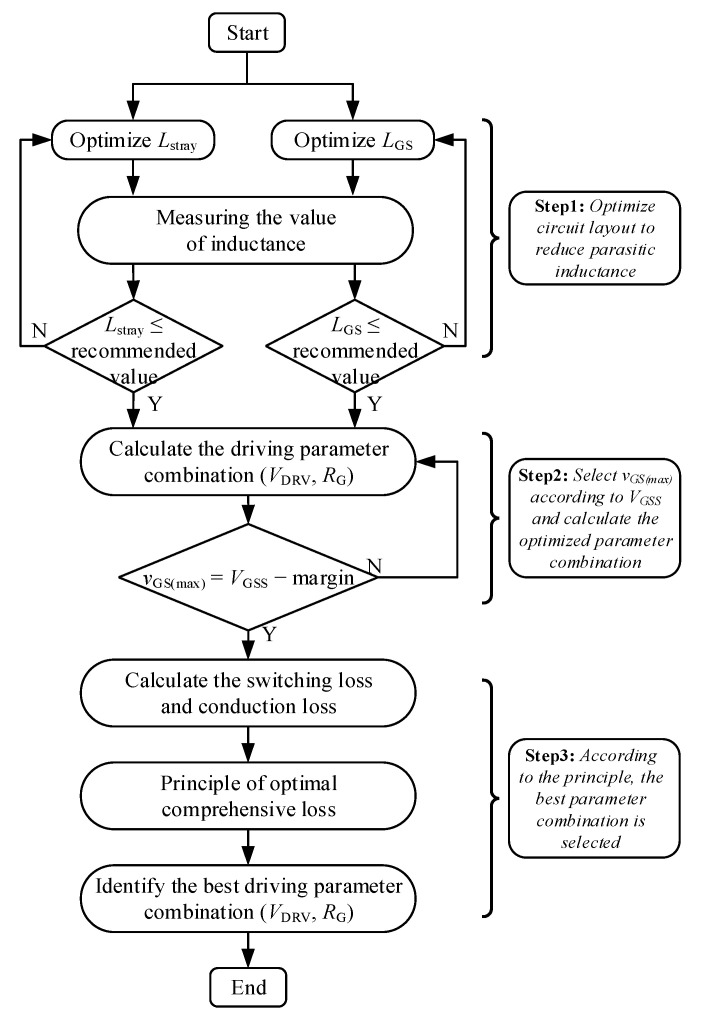
Methodology for optimizing driving parameters of SiC MOSFET.

**Figure 4 micromachines-14-00505-f004:**
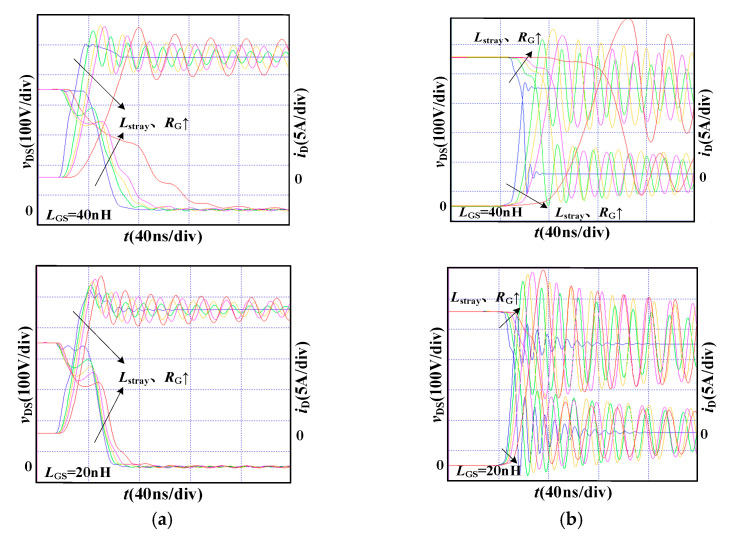
Switching waveforms under different stray inductance *L*_stray_. (**a**) turn-on process; (**b**) turn-off process.

**Figure 5 micromachines-14-00505-f005:**
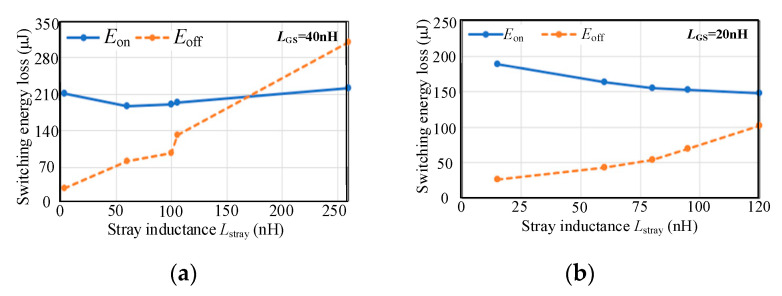
Switching energy loss under different *L*_stray_. (**a**) *L*_GS_ = 40 nH; (**b**) *L*_GS_ = 20 nH.

**Figure 6 micromachines-14-00505-f006:**
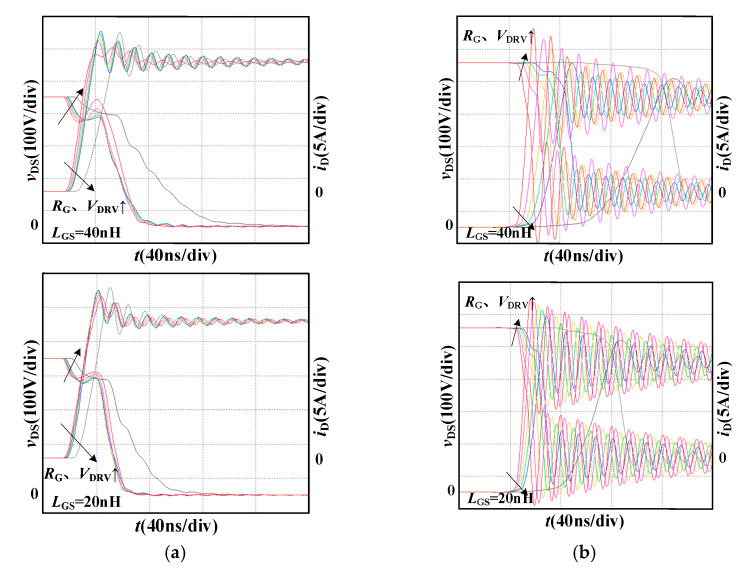
Switching waveforms under different driving parameters. (**a**) turn-on process; (**b**) turn-off process.

**Figure 7 micromachines-14-00505-f007:**
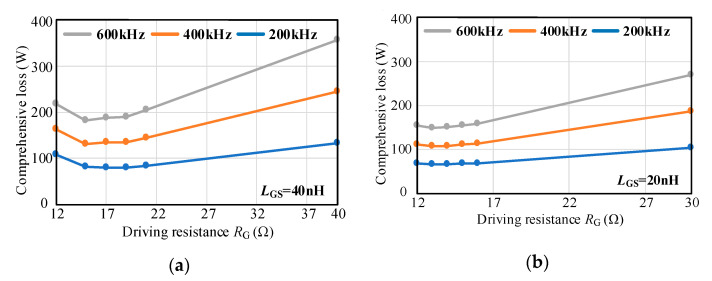
Comprehensive loss under different driving parameters. (**a**) *L*_GS_ = 40 nH; (**b**) *L*_GS_ = 20 nH.

**Figure 8 micromachines-14-00505-f008:**
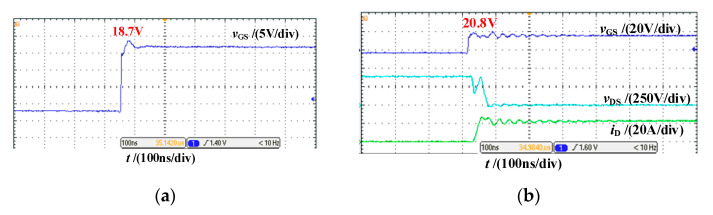
Gate-source voltage waveform under different operating conditions. (**a**) Power circuit not connected; (**b**) Bus voltage & load current: 400 V/20 A.

**Figure 9 micromachines-14-00505-f009:**
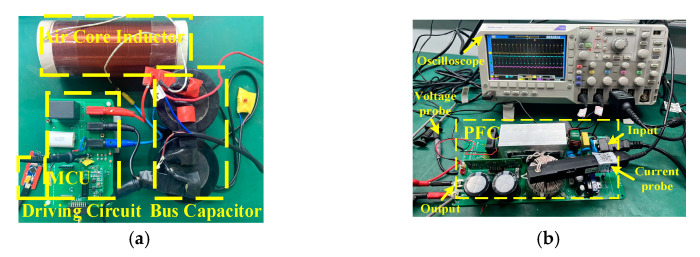
Experimental platform. (**a**) Double pulse test (DPT); (**b**) Multi pulse test (MPT).

**Figure 10 micromachines-14-00505-f010:**
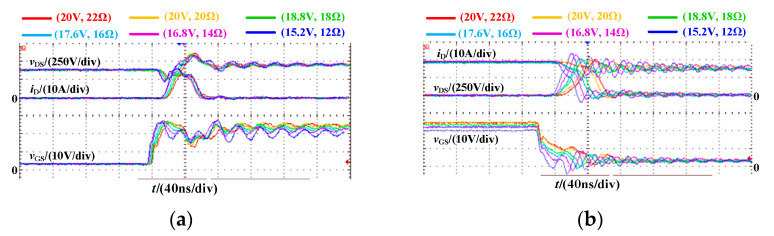
Switching waveform of SiC MOSFET under different driving parameter combinations in DPT. (**a**) Turn-on process; (**b**) Turn-off process.

**Figure 11 micromachines-14-00505-f011:**
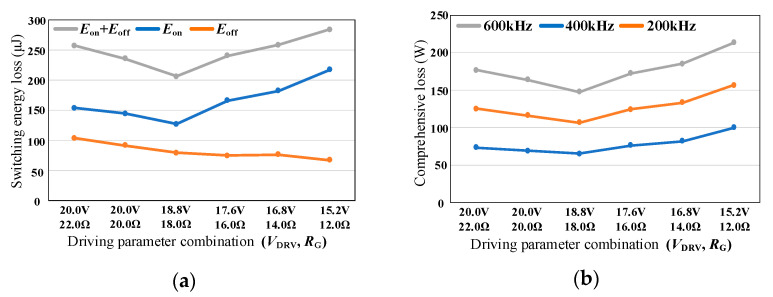
Switching energy loss and comprehensive loss under different driving parameter combinations. (**a**) switching energy loss; (**b**) comprehensive loss.

**Figure 12 micromachines-14-00505-f012:**
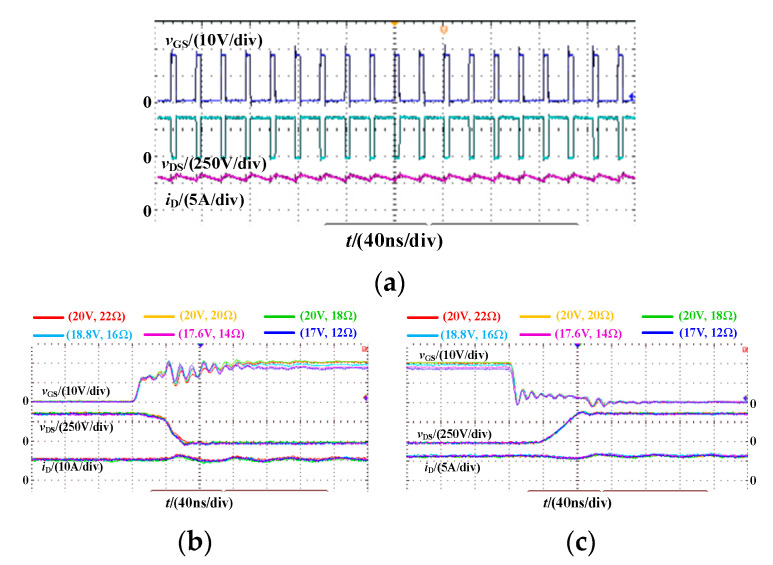
Multi pulse and switching waveform of SiC MOSFET. (**a**) Multi pulse; (**b**) Turn-on process; (**c**) Turn-off process.

**Figure 13 micromachines-14-00505-f013:**
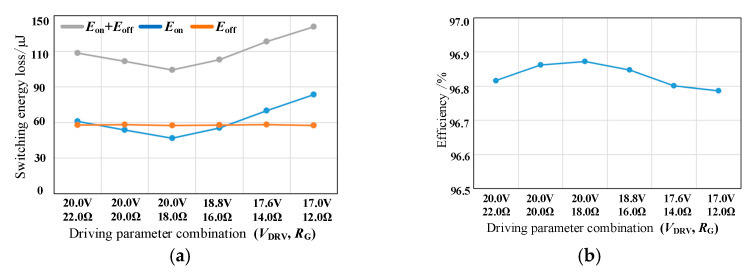
Loss and efficiency under different driving parameter combinations. (**a**) switching energy loss; (**b**) efficiency.

**Figure 14 micromachines-14-00505-f014:**
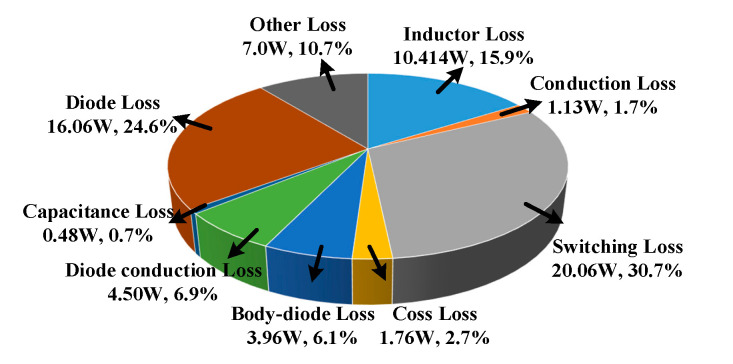
Power loss distribution of Boost PFC converter.

**Table 1 micromachines-14-00505-t001:** Circuit parameters and testing data.

*V*_DRV_/V	*L*_GS_/nH	*L*_stray_/nH	*R*_G_/Ω	*v*_DS(max)_/V	*E*_on_/μJ	*E*_off_/μJ
18	40	3	13.5	421	210.78	29.957
		60	17	569	187.78	81.901
		100	19	572	190.47	97.740
		105	21	582	194.02	131.39
		260	32	641	222.21	310.43
18	20	15	12	497	188.70	25.689
		60	13	608	163.65	42.792
		80	14	631	155.68	54.227
		95	15	640	152.85	69.766
		125	17	645	148.47	102.05

**Table 2 micromachines-14-00505-t002:** Circuit parameters and testing data.

*L*_GS_/nH	*V*_DRV_/V	*R*_G_/Ω	*v*_DS(max)_/V	*E*_on_/μJ	*E*_off_/μJ
40	15.8	12	639	234.88	39.152
	17.2	15	602	197.48	56.987
	18.0	17	569	187.78	81.901
	18.7	19	538	181.85	91.592
	19.5	21	520	183.85	119.06
	21.0	40	468	290.55	270.85
20	17.1	12	618	185.59	34.702
	18.0	13	608	163.65	42.792
	18.6	14	597	157.93	52.501
	19.0	15	583	154.74	63.532
	19.3	16	564	154.00	73.093
	21.0	30	479	219.30	198.44

**Table 3 micromachines-14-00505-t003:** Experimental test conditions.

Type	*V*_DC_/V	*I*_L_/A	*L*_GS_/nH	(*V*_DRV_, *R*_G_)/(V, Ω)
DPT	400	20	42.52	(15.2, 12), (16.8, 14), (17.6, 16), (18.8, 18), (20.0, 20), (20.0, 22).
MPT	400	8	19.34	(17.0, 12), (17.6, 14), (18.8, 16), (20.0, 18), (20.0, 20), (20.0, 22).

## Data Availability

Not applicable.

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
