# Peer review of "Parameters Design and Optimization of SiC MOSFET Driving Circuit with Consideration of Comprehensive Loss and Voltage Stress"

_micromachines, 2023, doi:10.3390/mi14030505_

Round 1

Reviewer 1 Report

Please introduce your reference:

  •  

Optimized Parameter Selection Method of Driving Circuit for SiC MOSFET

Haihong Qin;Sixuan Xie;Feifei Bu;Shishan Wang;Wenming Chen;Dafeng Fu

2021 IEEE Workshop on Wide Bandgap Power Devices and Applications in Asia (WiPDA Asia) Year: 2021 | Conference Paper | Publisher: IEEE      

Reviewer 2 Report

Parameters Design and Optimization of SiC MOSFET Driving Circuit with Consideration of Comprehensive Loss and Voltage Stress

My comments are as follows:

1. For a reader with some knowledge of the subject, most of the text and the figures are likely to come across as well-known material. I wonder whether this level of detail for the introductory part is necessary.

2. Authors should clearly state the limitations of the proposed method in real applications.

3. Authors should argue their choice of performance evaluation indicators.

4. The introduction section should be improved to provide adequate background information while also including recent relevant references. Following on from the previous comment, provide a summary of recent literature reviews, highlighting the gaps that need to be addressed and demonstrating how this study will address these gaps.

5. In the revised manuscript, the verification subsection between the analytical (simulation) solution and experimental should be highlighted, indicating the accuracy of the proposed design methodology.

6. Please comment on the values of the external gate resistance used in this paper, (from the data sheet it is recommended to use 0 ohms while figure 13 used 22 ohms!)

7. More analysis and discussions should be provided in case studies, e.g. Figures 7, 11, 13, and 14.

8.  How the authors calculate the losses, in Figure 14, is not clear.

9. Please comment on why the switching losses, in Figures 11 and 13, have U-shape (decrease then increase).

10. Please comment on the effect of increasing the switching frequency on the proposed methodology, e.g. switching losses.

11. The author's efforts are highly appreciated to conduct this research work. However, no comparison has been made with the in-use techniques to prove the supremacy of the proposed system. Furthermore, a lot of work has been done and can be found on the said problems with proposed. 

Reviewer 3 Report

               The article “Parameters Design and Optimization of SiC MOSFET Driving Circuit with Consideration of Comprehensive Loss and Voltage Stress” by Qin et al introduced a new mathematical model with optimization design method that obtains optimal parameters of the SiC MOSFET driving circuit with consideration of previously often ignored parasitic parameters. The authors did a great job introducing the establishment of model with strong theory support. Meanwhile, parameter optimization design method follows clear and coherent logical paths. Experimental verification further proves the accuracy and correctness of this optimization method by going through “Double pulse test” and “multi-pulse test”.  This paper has potential great impact in semiconductor industry by offering a promising method to improve the SiC MOSFET performances. The illustration is solid and clear.

Reviewer 4 Report

Switching Oscillations have been investigated by other groups, you can refer to the paper " Modeling and Analysis of SiC MOSFET Switching Oscillations, IEEE JOURNAL OF EMERGING AND SELECTED TOPICS IN POWER ELECTRONICS, VOL. 4, NO. 3, SEPTEMBER 2016"

Round 2

Reviewer 1 Report

It seems to me that the manuscript could be published.

Reviewer 2 Report

There has been a good revision of the manuscript by the authors. Therefore, the manuscript can be accepted as it is.